# Evaluation of the Impact of Coinfection and Superinfection on Chikungunya and Mayaro Viruses’ Replication in *Aedes aegypti*

**DOI:** 10.3390/microorganisms13092165

**Published:** 2025-09-17

**Authors:** Maria Eduarda dos Santos Pereira de Oliveira, Larissa Krokovsky, Maria Júlia Brito Couto, Duschinka Ribeiro Duarte Guedes, George Tadeu Nunes Diniz, Constância Flávia Junqueira Ayres, Marcelo Henrique Santos Paiva

**Affiliations:** 1Department of Entomology, Instituto Aggeu Magalhães, Fundação Oswaldo Cruz, Av. Professor Moraes Rego, S/N, Campus da UFPE, Cidade Universitária, Recife 50740-465, Brazil; mariaeduardaspoliveira@gmail.com (M.E.d.S.P.d.O.); julia.britoc@ufpe.br (M.J.B.C.); duschinka.guedes@fiocruz.br (D.R.D.G.); constancia.ayres@fiocruz.br (C.F.J.A.); 2Department of Mathematics and Science, Brock University, 1812 Sir Isaac Brock Way, St. Catharines, ON L2S 3A1, Canada; lkrokovsky@gmail.com; 3Laboratory of Quantitative Methods, Instituto Aggeu Magalhães, Fundação Oswaldo Cruz, Av. Professor Moraes Rego, S/N, Campus da UFPE, Cidade Universitária, Recife 50740-465, Brazil; george.diniz@fiocruz.br; 4Núcleo de Ciências da Vida, Centro Acadêmico do Agreste, Universidade Federal de Pernambuco (UFPE), Rodovia BR-104, km 59-Nova Caruaru, Caruaru 55002-970, Brazil

**Keywords:** *Aedes aegypti*, alphavirus, replication, cocirculation

## Abstract

The simultaneous circulation of multiple arboviruses, often driven by (re)emergence events, poses challenges to public health systems. In Brazil, the co-circulation of Dengue virus (DENV), Zika virus (ZIKV), Chikungunya virus (CHIKV), and Oropouche virus (OROV), together with the potential urban emergence of Mayaro virus (MAYV), underscores the importance of understanding interactions among these pathogens within their vectors. This study investigated the effects of CHIKV and MAYV coinfection and superinfection on replication dynamics in *Aedes aegypti*. Mosquitoes were experimentally exposed to CHIKV and MAYV through artificial blood meals under coinfection and superinfection conditions. Infection (IR), dissemination (DR), and transmission (TR) rates, as well as viral loads, were quantified by quantitative reverse transcription PCR (qRT-PCR). To confirm viral replication and assess cytopathic effects, positive saliva samples were inoculated in Vero cells, followed by serial passages and plaque assays for viral titration. The results showed that *Ae. aegypti* is capable of transmitting both CHIKV and MAYV concurrently during coinfection. However, in superinfection scenarios, prior infection with either virus significantly reduced the transmission efficiency of the subsequently acquired virus, indicating viral interference at the replication level. These findings underscore the complexity of arboviral interactions within vectors and highlight their potential implications for transmission dynamics. Continuous entomo-virological surveillance and targeted research are essential for anticipating and mitigating the impact of arboviral co-circulation in endemic regions.

## 1. Introduction

The emergence and re-emergence of arthropod-borne viruses (arboviruses) are increasingly linked to intensified interactions among viruses, vectors, and human hosts [1]. This epidemiological scenario is influenced by a range of factors such as globalization, the anthropogenic disruption of natural ecosystems, climate change, the geographic expansion of competent vectors, and inadequate urban sanitation infrastructure [2,3]. These conditions collectively promote elevated vector densities, a greater availability of susceptible hosts, and enhanced environmental suitability for the circulation, persistence, and dissemination of multiple arboviruses [1,2,3]. In Brazil, the concurrent transmission of Dengue virus (DENV), Zika virus (ZIKV), and Chikungunya virus (CHIKV) culminated in a triple epidemic during 2015–2016 [4]. Since then, these arboviruses have become endemic in urban areas, with *Aedes aegypti* serving as the primary mosquito vector and humans as the primary vertebrate host [5]. Arboviral co-circulation is shaped by overlapping biological, ecological, and socioeconomic factors, which collectively drive synergistic epidemiological patterns, particularly with respect to seasonal transmission and incidence rates [3,6,7,8]. More recently, Mayaro virus (MAYV), an Alphavirus historically confined to sylvatic environments, has emerged as a pathogen of growing concern due to its potential for urban establishment and epidemic spread [9,10,11]. Confirmed autochthonous cases have now been documented in the north, midwest, southeast, and northeast regions of Brazil [12].

One of the key determinants for the establishment of these viruses, including MAYV, in urban environments is the presence of competent mosquito vectors [13,14,15]. In its sylvatic transmission cycle, MAYV is maintained primarily between non-human primates and other mammals as vertebrate hosts, with *Haemagogus* spp. mosquitoes serving as the main vectors [9,11]. However, entomological surveillance studies in Brazil have reported natural infections of *Ae. aegypti* with MAYV in the states of Mato Grosso [16] and Goiás [17]. These findings, together with the recurrent detection of MAYV cases near major tropical urban centers, the widespread presence of *Ae. aegypti* as the dominant urban vector, and the availability of susceptible human hosts, underscore the potential for MAYV to establish a sustained urban transmission cycle [18].

Interactions among distinct arboviruses within a single mosquito are critical for understanding transmission dynamics. Such interactions occur primarily through two mechanisms: coinfection, in which a mosquito acquires multiple viruses during a single viremic blood meal, and superinfection, in which viruses are acquired sequentially from different hosts [19,20]. These dual-infection scenarios can generate diverse outcomes, including viral amplification, inhibition, competitive interference, or neutral interactions [3,21]. In natural settings, superinfection is considered more likely than coinfection, as mosquitoes typically feed on multiple viremic hosts rather than a single host concurrently infected with multiple viruses [3,22,23]. Superinfection frequently leads to viral interference or superinfection exclusion, a phenomenon in which a pre-existing infection impairs or prevents the replication of a subsequently acquired virus within the same host or cell [24,25,26]. This process has been demonstrated in interactions between closely related viruses, such as Yellow Fever virus (YFV) and Dengue virus serotype 2 (DENV-2) [27], as well as between Chikungunya virus (CHIKV) and MAYV [22]. Proposed mechanisms of interference include competition for cellular resources, inhibition by virus-derived molecules, or the activation of virus-specific host defense pathways [26]. Moreover, viral interference has also been reported to be observed between viruses belonging to different families, such as MAYV and Zika virus (ZIKV) [21]. Superinfection exclusion may confer evolutionary advantages by minimizing interviral competition within the host and enhancing viral fitness [26].

In the context of arboviral co-circulation in Brazil and given the high abundance of *Ae. aegypti*, we investigated the effects of coinfection and superinfection on the replication dynamics of two medically relevant alphaviruses, CHIKV and MAYV. Using a laboratory-reared colony of *Ae. aegypti*, we conducted artificial blood-feeding experiments to model both infection scenarios and performed comparative analyses with single infections to assess the nature and magnitude of viral interactions.

## 2. Materials and Methods

### 2.1. Viral Stock

Stocks of Chikungunya virus (CHIKV BRPE408/2016) and Mayaro virus (MAYV/BR/Sinop/H307/2015; GenBank accession MH513597) were generously provided by Dr. Marli Tenório (Department of Virology, LAVITE, Aggeu Magalhães Institute) and Dr. Roberta Bronzoni (Federal University of Mato Grosso, UFMT-Sinop), respectively. Viral propagation and titration were carried out in Vero cell monolayers, and viral titers were determined using plaque assays, expressed as plaque-forming units per milliliter (PFU/mL), according to the protocol described by Krokovsky et al. (2023) [15].

### 2.2. The Aedes aegypti Colony

The *Aedes aegypti* colony (RecLab) used in this study originated from a natural population collected in Recife, Pernambuco, Brazil [28]. Mosquitoes were maintained under standard insectary conditions: at 26 ± 2 °C, with a relative humidity of 65–85%, and a 14:12 h light/dark photoperiod, until adult emergence, after which they were allocated to experimental groups. This study did not require approval from an institutional ethics committee, as it exclusively involved laboratory-reared *Ae. aegypti* mosquitoes and did not include experiments with vertebrate animals or human subjects.

### 2.3. Aedes aegypti Artificial Blood Feeding

All experimental procedures were conducted under Biosafety Level 2 (BSL-2) containment. Female mosquitoes (7–10 days old) from the RecLab colony were placed into plastic cages containing 100 females and 20 males per group and maintained under controlled environmental conditions. To increase feeding propensity, mosquitoes were deprived of sugar 24 h prior to the feeding assays. Viral stocks of MAYV and CHIKV were propagated in Vero cells and harvested 24 h post-inoculation at a multiplicity of infection (MOI) of 0.1. Supernatants from uninfected Vero cells served as negative controls.

For artificial blood meals, freshly harvested viral supernatants were mixed at a 1:1 ratio with defibrinated rabbit blood and supplemented with adenosine triphosphate (ATP) at a final concentration of 3 mM to stimulate feeding. Mosquitoes were permitted to feed for up to one hour using a membrane feeding system consisting of Petri dishes sealed with a triple layer of Parafilm, as previously described by Guedes et al. (2017) [29]. Following blood feeding, fully engorged females were anesthetized on ice, transferred to clean cages, and maintained for 13 days post-exposure (dpe). Each experimental group received two blood meals in accordance with the specific protocols for single-infection, coinfection, and superinfection assays, as illustrated in the experimental design (Figure 1).

In the single-infection assays, mosquitoes were offered an initial infectious blood meal on day 0 containing either MAYV or CHIKV, followed by a noninfectious blood meal at 6 days post-exposure (dpe). In the coinfection group, both viruses were simultaneously administered in a single blood meal at equal titers on day 0, with a subsequent non-infectious blood meal provided at 6 dpe. For the superinfection groups, mosquitoes were sequentially exposed to MAYV and CHIKV through two distinct infectious blood meals administered six days apart. The first blood meal, on day 0, contained only one virus, while the second, on 6 dpe, contained the heterologous virus, enabling evaluation of interviral interactions from the second exposure onward. In the CHIKV superinfection group, mosquitoes were initially infected with MAYV and subsequently exposed to CHIKV. Conversely, in the MAYV superinfection group, mosquitoes were first exposed to CHIKV and later to MAYV at 6 dpe. Viral titers in all infectious blood meals were quantified and standardized to ensure consistency across experimental conditions. To approximate upper-range human viremia for CHIKV and to follow standard vector competence practice for alphaviruses (including MAYV), we targeted blood meal titers of ~10^7^ PFU/mL; this level is within documented acute human viremia for CHIKV and, although above typical human/NHP viremia reported for MAYV, is widely used to ensure consistent midgut infection and enable comparisons across single-, co-, and superinfection conditions (Table 1) [30,31,32,33,34].

### 2.4. Sample Collection

Single-infection, coinfection, and superinfection experiments were performed in duplicate, resulting in a total of *n* = 10 samples per group (5 per replicate) for each tissue type (midguts, carcasses, and saliva) in the single-infection and negative control groups. For the coinfection and superinfection groups, *n* = 30 samples per group (15 per replicate) were collected at each time point.

Midguts and carcasses were dissected from female mosquitoes at 6 days post-exposure (dpe), immediately prior to the second blood meal, and again at 12 dpe (Figure 1). Mosquitoes were cold-anesthetized at –20 °C, rinsed individually in 70% ethanol followed by sterile distilled water, and transferred to 1× phosphate-buffered saline (PBS; Gibco, Waltham, MA, USA). Dissections were performed in Petri dishes containing 1× PBS using entomological pins and fine forceps under a binocular stereomicroscope (LABOMED, Los Angeles, CA, USA). Each tissue was placed into a 1.5 mL DNase/RNase-free microtube containing 300 μL of mosquito diluent (Leibovitz’s L-15 medium, Gibco) and stored at −80 °C until further processing.

Saliva samples were collected at 7 and 13 dpe (Figure 1). To ensure experimental feasibility, saliva was obtained from individuals not used for dissection; however, all individuals originated from the same laboratory colony to maintain consistency in physiological response. Mosquitoes were cold-anesthetized at −20 °C, and their legs and wings were removed to facilitate salivation. The proboscis was inserted into a 100 μL pipette tip containing 10 μL of 1× PBS and left for 30 min. The forced salivation procedure was based on the method described by Heitmann et al. (2018), with minor modifications [35]. Following salivation, the PBS solution was expelled into a 1.5 mL DNase/RNase-free microtube containing an additional 10 μL of 1× PBS and stored at –80 °C for subsequent qRT-PCR analysis. Carcasses of mosquitoes yielding positive saliva samples were also retained for further processing.

### 2.5. RNA Isolation and qRT-PCR

Collected tissues were individually homogenized as previously described by Bar-bosa et al. (2016) [36], and RNA was extracted using the TRIzol^®^ protocol (Invitrogen, Carlsbad, CA, USA), as outlined by Guedes et al. (2017) [29]. RNA extracted from midguts and carcasses and saliva samples were used as templates for qRT-PCR reactions, which were performed with the QuantiNova Probe RT-PCR Kit (Qiagen, Hilden, North Rhine-Westphalia, Germany) in a final volume of 10 μL (3.5 μL of RNA). Reaction mixtures included final concentrations of 1X for QuantiNova Probe RT-PCR Master Mix, QuantiNova ROX, and QuantiNova RT Mix; 800 nM of CHIKV primers; 1200 nM of MAYV primers; 200 nM of CHIKV probe; and 300 nM of MAYV probe. Primer and probe sequences for CHIKV and MAYV were previously reported by Lanciotti et al. (2007) [37] and Naveca et al. (2017) [38], respectively.

Amplification reactions were conducted on a QuantStudio 5 System (Applied BioSystems, Norwalk, CT, USA) under the following conditions: 45 °C for 15 min, 95 °C for 5 min, followed by 45 cycles of 95 °C for 5 s, and 60 °C for 45 s. Each sample was analyzed in duplicate, with negative controls consisting of no-template controls (NTCs), and positive controls consisting of standard curves for each virus (MAYV and CHIKV), synthesized as described in Krokovsky et al. (2022) [39]. qRT-PCR data were analyzed using QuantStudio Design and Analysis Software v. 1.3.1 with automatic threshold and baseline. Samples with Cq (cycle quantification) values ≤ 38.5 for both MAYV and CHIKV were considered positive.

### 2.6. Virus Propagation and Titration

Saliva samples that tested positive by qRT-PCR were inoculated onto Vero cell monolayers to evaluate the presence of cytopathic effects (CPEs) as indicators of active viral replication and associated cell mortality. Vero cells were seeded in 96-well plates, and 4 μL of each saliva sample was inoculated into individual wells. After a 1 h adsorption period at 37 °C, 50 μL of minimum essential medium supplemented with 2% fetal bovine serum (2% MEM) was added to each well. Infected cultures were monitored daily for up to 96 h post-infection (hpi) for the appearance of CPEs. Upon detection of a CPE, culture supernatants were collected and stored at –80 °C. These supernatants were subsequently used to infect fresh Vero cell monolayers under the same conditions for up to three serial passages in order to confirm and amplify infectious viral particles. After each passage, supernatants were harvested and subjected to plaque assays to quantify viral titers, expressed as plaque-forming units per milliliter (PFU/mL).

### 2.7. Data Analysis

Infection rate (IR), dissemination rate (DR), and transmission rate (TR) were calculated at designated time points as follows: IR = number of positive midguts/total midguts analyzed; DR = number of positive carcasses/number of positive midguts; and TR = number of positive saliva samples/total saliva samples analyzed. IR and DR were evaluated at 6- and 12-days post-exposure (dpe), whereas TR was assessed at 7 and 13 dpe. Categorical variables were analyzed using logistic regression, Pearson’s chi-square test, or Fisher’s exact test, depending on data distribution and sample size. To verify the assumption of homogeneity of variance in viral load data (expressed as RNA copies/mL), Bartlett’s test was applied. When the assumption was not met, non-parametric analyses were performed using the Kruskal–Wallis test, followed by Fisher’s least significant difference (LSD) post hoc test for pairwise comparisons among experimental groups and time points. A significance level of *p* < 0.05 was adopted for all statistical tests. Analyses were performed using R statistical software (R Core Team, 2023).

## 3. Results

### 3.1. Single-Infection and Coinfection Assays

To facilitate comparative analysis, data from the single-infection and coinfection groups are presented in the same graphical panels. In the MAYV single-infected group, the infection rate (IR) was 100% at 6 days post-exposure (dpe) and 90% at 12 dpe (Figure 2A), while the dissemination rate (DR) remained at 100% for both time points (Figure 2C). Mean viral loads in midgut samples were 1.3 × 10^9^ RNA copies/mL at 6 dpe and 1.9 × 10^10^ RNA copies/mL at 12 dpe (Figure 2B). In carcass samples, average viral loads were higher than in midguts, reaching 1 × 10^11^ RNA copies/mL at 6 dpe and 6.8 × 10^11^ RNA copies/mL at 12 dpe, representing an increase of approximately two log_10_ units (Figure 2D). For the CHIKV single-infected group, both IR and DR were 100% at 6 and 12 dpe (Figure 3A,C). Mean midgut viral loads were 1 × 10^12^ RNA copies/mL at 6 dpe and 1.8 × 10^12^ RNA copies/mL at 12 dpe (Figure 3B). Viral loads in carcasses were approximately one log_10_ unit higher than those in midguts, with mean values of 4.1 × 10^13^ RNA copies/mL at 6 dpe and 1.5 × 10^13^ RNA copies/mL at 12 dpe (Figure 3D).

Coinfection with CHIKV and MAYV did not produce statistically significant differences in IR when compared to the respective single-infection controls. Within the coinfection group, CHIKV showed a slightly higher IR (96.7% at 6 dpe and 93.3% at 12 dpe) compared to MAYV, which remained at 93.2% across both time points (Figure 3A). However, CHIKV displayed significantly higher viral loads than MAYV. In the midgut, CHIKV reached average titers of 4.6 × 10^11^ RNA copies/mL at 6 dpe and 8.8 × 10^11^ RNA copies/mL at 12 dpe, whereas MAYV titers were 1.5 × 10^9^ and 1.5 × 10^10^ RNA copies/mL at 6 and 12 dpe, respectively (Figure 2B and Figure 3B). Both viruses achieved 100% dissemination in the coinfection group (Figure 2C and Figure 3C). Nevertheless, CHIKV also showed higher viral loads in carcass samples, with mean values of 6.5 × 10^12^ RNA copies/mL at 6 dpe and 1.3 × 10^13^ RNA copies/mL at 12 dpe, compared with MAYV titers of 1.5 × 10^10^ and 1.1 × 10^11^ RNA copies/mL at the respective time points (Figure 2D and Figure 3D).

In the MAYV single-infection group, the transmission rate (TR) was 30% at 7 days post-exposure (dpe), decreasing to 10% at 13 dpe (Figure 4A). Corresponding mean viral loads in saliva samples were 3.6 × 10^6^ RNA copies/mL at 7 dpe and 8.6 × 10^3^ RNA copies/mL at 13 dpe (Figure 4B). In contrast, CHIKV single-infected mosquitoes showed no detectable transmission at either time point, with no saliva samples testing positive (Figure 4C,D). In the coinfection group, MAYV exhibited a TR of 16.7% at both 7 and 13 dpe (Figure 4A). Mean viral loads in saliva were 1.5 × 10^5^ RNA copies/mL at 7 dpe and 1.02 × 10^7^ RNA copies/mL at 13 dpe (Figure 4B). For CHIKV, a TR of 7.2% was observed at 7 dpe (Figure 4C), with a corresponding mean saliva viral load of 5.7 × 10^8^ RNA copies/mL (Figure 4D). No CHIKV-positive saliva samples were detected at 13 dpe in the coinfection group.

In the MAYV single-infection group, all qRT-PCR-positive saliva samples (3/3) induced cytopathic effects (CPEs) following serial passage in Vero cells, confirming active viral replication. The corresponding carcasses also tested positive, with a mean viral load of 1.1 × 10^12^ RNA copies/mL. In the coinfection group, 40% (4/10) of the MAYV-positive saliva samples induced CPEs, and all associated carcasses were positive, with an average viral load of 4.1 × 10^11^ RNA copies/mL. For CHIKV in the coinfection group, both qRT-PCR-positive saliva samples (2/2) produced CPEs in Vero cells, and all corresponding carcasses tested positive with a mean viral load of 1.1 × 10^14^ RNA copies/mL. Despite the observed cytopathic activity in these samples, no detectable viral titers were obtained by plaque assay. These findings should be interpreted with caution, as the number of positive saliva samples was limited in all groups, even when positivity rates reached 100%.

### 3.2. Superinfection Assay

In the CHIKV superinfection group, in which mosquitoes were first exposed to MAYV and subsequently to CHIKV, MAYV infection rates (IRs) were 70% at 6 days post-exposure (dpe) and increased to 93.3% at 12 dpe (Figure 5A). The CHIKV IR reached 83.3% at 12 dpe (Figure 5E). MAYV dissemination rates (DRs) were 83.3% at 6 dpe and 93.1% at 12 dpe (Figure 5C), whereas the CHIKV DR was 85.7% at 12 dpe (Figure 5G). The mean MAYV RNA loads in midguts were 1.7 × 10^11^ copies/mL at 6 dpe and 4.1 × 10^11^ copies/mL at 12 dpe (Figure 5B). In carcasses, viral loads averaged 3.6 × 10^11^ copies/mL at 6 dpe and 1.6 × 10^13^ copies/mL at 12 dpe (Figure 5D). For CHIKV, the mean RNA loads in midguts and carcasses at 12 dpe were 2.1 × 10^11^ and 2.0 × 10^12^ copies/mL, respectively (Figure 5F,H).

In the MAYV superinfection group—in which CHIKV was administered first, followed by MAYV—theCHIKV IRs were 73.3% at 6 dpe and 76.7% at 12 dpe (Figure 5E), while the MAYV IR at 12 dpe was 73.3% (Figure 5A). The DR for CHIKV was 95.5% at 6 dpe but declined to 61.0% at 12 dpe (Figure 5G). For MAYV, the DR at 12 dpe was 71.4% (Figure 5C). The CHIKV RNA loads in midguts averaged 1.3 × 10^11^ copies/mL at 6 dpe and 2.4 × 10^11^ copies/mL at 12 dpe (Figure 5F), while the carcass viral loads were 1.8 × 10^12^ and 2.1 × 10^12^ copies/mL at 6 and 12 dpe, respectively (Figure 5H). For MAYV, the midgut viral loads at 12 dpe averaged 3.1 × 10^11^ copies/mL, and the carcass loads reached 1.8 × 10^12^ copies/mL (Figure 5B,D).

In the CHIKV superinfection assay, MAYV was detected in 6.7% of saliva samples at 7 dpe (Figure 6A), with a mean viral load of 9 × 10^10^ RNA copies/mL (Figure 6B). In the reciprocal condition, mosquitoes superinfected with MAYV exhibited a 3.3% CHIKV positivity rate in saliva samples at 13 dpe (Figure 6C), with a mean viral load of 3.9 × 10^10^ RNA copies/mL (Figure 6D).

In the CHIKV superinfection group, neither of the two MAYV-positive saliva samples induced CPEs in Vero cells during any of the three serial passages. Nevertheless, viral RNA was detected in the corresponding carcasses, with a mean viral load of 1.8 × 10^11^ RNA copies/mL, confirming systemic infection. In contrast, the CHIKV-positive saliva sample obtained from the MAYV superinfection group produced CPEs following inoculation, and the corresponding carcass also tested positive for CHIKV RNA, with a viral load of 3.3 × 10^14^ RNA copies/mL. Despite the presence of viral RNA in all samples and the occurrence of CPEs in some cases, none yielded detectable viral titers by plaque assay. This discrepancy between RT-qPCR and plaque assays on Vero cells likely reflects the presence of non-infectious or defective viral particles, as reported in other arbovirus studies, and should therefore be interpreted with caution.

## 4. Discussion

The current global epidemiological landscape is characterized by the frequent emergence and (re)emergence of arboviruses and their rapid geographic expansion [1]. This trend has contributed to the simultaneous circulation of multiple arboviral pathogens within specific regions [40]. In Brazil, in addition to endemic arboviruses such as Dengue virus (DENV), Zika virus (ZIKV), Chikungunya virus (CHIKV), and Yellow Fever virus (YFV), Oropouche virus (OROV) has recently emerged as a significant public health concern, with over 4000 confirmed cases reported [41]. Within this context, Mayaro virus (MAYV) is of particular concern due to its increasing detection in peri-urban and urban areas of Brazil, which currently report the highest number of human MAYV infections worldwide [9,10,11,12,18]. The convergence of competent mosquito vectors, susceptible human hosts, and favorable ecological and environmental conditions underscores the potential for the concurrent transmission of multiple arboviruses [5]. Understanding the interactions between co-circulating viruses within a shared mosquito vector is therefore critical for anticipating and managing arboviral transmission dynamics [22]. Despite its importance, this area of research remains underexplored and is marked by significant knowledge gaps [21]. In light of this context, the present study investigated the effects of coinfection and superinfection on the replication dynamics of CHIKV and MAYV in *Ae. aegypti*, with comparisons to their respective single infections.

In coinfection scenarios, both CHIKV and MAYV replicated at levels comparable to their respective single infections, suggesting limited or no interference with viral replication or transmissibility. These findings are consistent with previous studies evaluating the vector competence of *Ae. aegypti*, which, using similar sample sizes (10–40 mosquitoes per time point), have demonstrated the ability of this species to become simultaneously infected with and transmit multiple arboviruses from a single infectious blood meal. Documented combinations include DENV, ZIKV, and CHIKV [42,43], ZIKV-MAYV [21], and CHIKV–MAYV [22,44], aligning with the results observed in the present study. Collectively, these studies support the hypothesis that coinfection does not substantially impair viral amplification within the vector [3,43,45]. Moreover, Le Coupanec et al. (2017) showed that mixed infections with CHIKV and DENV can enhance viral replication in *Ae. aegypti*, indicating that virus–virus interactions may be context-dependent [46]. Silva et al. (2025) observed lower viral loads for each virus in coinfected Vero cells compared with their respective single infections [47]. Similarly, in our study, MAYV showed reduced viral titers in coinfection settings compared to single infections, whereas CHIKV titers were similar or slightly lower. However, these differences may reflect experimental variation rather than true biological effects. Environmental factors, such as temperature, can further shape viral dynamics. For example, Terradas et al. (2024) found similar infection rates for MAYV and DENV in coinfection experiments in *Ae. aegypti* and cell lines (Vero, C6/36, and *Ae. aegypti* cells), but DENV exhibited higher dissemination, and both viruses showed increased transmission under elevated temperatures [48].

Notably, our superinfection experiments revealed evidence of viral interference affecting transmission. In the CHIKV-superinfected group, in which MAYV was administered first, only MAYV RNA was detected in saliva samples collected at 7 dpe, suggesting a possible delay in CHIKV replication and/or dissemination. This may reflect insufficient time for the secondary virus to overcome tissue barriers and reach the salivary glands [49]. Conversely, in the MAYV-superinfected group, in which CHIKV was administered first, only CHIKV RNA was detected in saliva at 13 dpe. These findings are partially consistent with those of Kantor et al. (2019), who reported viral interference in *Ae. aegypti* superinfected with CHIKV and MAYV; interference was observed when CHIKV followed MAYV, but not in the reverse order [22]. Silva et al. (2025) demonstrated using Vero cells in vitro that superinfections led to significant differences in progeny production for each virus, with a clear advantage for the virus introduced first (MAYV over CHIKV and CHIKV over MAYV). Moreover, viruses that infected cells first produced progeny titers substantially higher than those observed in their respective single infections [47]. Interestingly, in our study analyses of midguts and carcasses, MAYV exhibited higher viral titers in superinfection settings compared to single infections, although this may reflect experimental variation rather than true biological enhancement. In contrast, CHIKV titers were similar or lower in both midguts and carcasses. These findings should be interpreted with caution. Although some evidence of transmission interference was observed, the overall transmission rates (6.7% and 3.3%, respectively) were low, limiting the strength of these conclusions regarding superinfection exclusion or inhibition of viral replication. These results underscore the need for additional experiments with larger sample sizes to confirm the observed trends. Nevertheless, by applying a side-by-side design under identical conditions, our study contributes additional evidence that arboviral interactions in *Ae. aegypti* are complex and context-dependent and highlights critical areas for future investigation.

Beyond molecular analyses, we conducted cell-based assays using saliva samples that tested positive by qRT-PCR in order to further investigate virus–cell interactions, an essential component for understanding arboviral dynamics within the vector and assessing transmission potential [26]. All corresponding mosquito carcasses from saliva-positive individuals tested positive across experimental groups, confirming the successful systemic dissemination of the viruses. However, when these saliva samples were used to inoculate Vero cells, infection rates were variable and generally low. Methodological limitations may have influenced these outcomes. The forced salivation technique, although widely regarded as the gold standard for assessing vector transmission capacity without the use of animal models, has inherent drawbacks, including the lack of standardization across studies and the inability to quantify the volume of saliva expelled [50]. These factors may contribute to variability and a potential underestimation of transmission rates. Alternatively, the low transmission rates observed may reflect biological barriers within the *Ae. aegypti* RecLab colony, specifically at the level of the salivary glands, which could restrict the release of infectious CHIKV and MAYV particles despite systemic viral dissemination [45,51,52]. Previous studies likewise reported low transmission rates for CHIKV and MAYV in *Ae. aegypti*. For example, Göertz et al. (2017) reported a CHIKV transmission rate of 21.2% at 14 dpe in single-infected mosquitoes using high initial titers (~10^7^ TCID_50_/mL) and in vitro infection assays [45]. In the same study, coinfection with ZIKV significantly reduced CHIKV transmission (14.6%) compared to ZIKV (72.9%), despite identical inoculum titers. Wiggins et al. (2018) similarly observed low transmission rates for MAYV, with values of 10% at 6 dpe and 23% at 12 dpe in single-infected mosquitoes exposed to a viral titer of 10^10^ PFU/mL, with transmission assessed by direct RNA extraction followed by molecular detection [53]. These observations and the inherent limitations of in vitro assays highlight the need to complement vector competence studies with animal models that allow for in vivo assessments of transmission. Such models can facilitate the evaluation of histopathological changes and viral tropism across mosquito tissues, thereby providing a more comprehensive understanding of arbovirus–vector interactions, as emphasized by Krokovsky et al. (2023) [15].

Previous studies have identified several factors that may contribute to viral interference, inhibition, or exclusion during superinfection scenarios [27]. A primary mechanism involves the activation of the vector’s innate immune system following initial infection, which can hinder the replication of a subsequently acquired arbovirus [3,27]. Additional contributing factors include microbial competition within the vector’s microbiome, the depletion of cellular resources essential for viral replication, the reduced availability or affinity of cell surface receptors for virion binding during the second infection, and the inhibition of nucleocapsid uncoating specific to the secondary virus [25,54,55]. Consistent with these mechanisms, Boussier et al. (2020) demonstrated that CHIKV induces superinfection exclusion even under low-viral-load conditions, using single-cell technology in vitro. In their study, CHIKV was tested as the secondary virus, along with Sindbis virus, a related alphavirus, and influenza A, an unrelated RNA virus. By dissecting the viral life cycle, the authors showed that CHIKV-mediated exclusion occurs at the replication stage rather than during virus binding [56]. In the case of alphaviruses, superinfection exclusion appears to be partially mediated by the nonstructural protein nsP2 of the primary infecting virus. Following the proteolytic processing of nonstructural polyproteins (e.g., nsP3/nsP4, nsP1/nsP2, and nsP2/nsP3), free nsP2 interferes with the replication of secondary viruses by preventing the formation of new viral replication complexes within the already infected host cell [57,58]. Supporting this, Reitmayer et al. (2023) demonstrated that nsP2 expression in *Ae. aegypti*-derived Aag2 cells and in transgenic mosquitoes significantly reduced the replication and transmission of Sindbis virus (SINV) and CHIKV. They further observed that the cytoplasmic localization of nsP2 effectively blocked viral replication complex assembly [59]. Interestingly, Boussier et al. (2020) reported nsP2 had no role in CHIKV, making it unique among characterized alphaviruses. In their study, individual CHIKV proteins were tested in cell lines, but none alone could reproduce complete superinfection exclusion [56]. Despite these mechanistic insights, the phenomenon of superinfection exclusion remains context-dependent and has not been universally observed. Some studies have reported a lack of interference in superinfection models. For example, Kuwata et al. (2015) showed that pre-existing infection with *Culex* flavivirus (CxFV), an insect-specific flavivirus isolated from *Culex* spp., did not inhibit the replication of Japanese encephalitis virus (JEV) or dengue virus (DENV) in cultured CTR cells. The authors of [60] demonstrated in vitro that a pre-existing infection with CxFV (an insect-specific flavivirus initially isolated from *Culex* spp.) did not interfere with Japanese Encephalitis virus (JEV) or DENV infection at the cellular (CTR cells) level. Similarly, Kent et al. (2010) found no impact of prior CxFV exposure on West Nile virus (WNV) replication in C6/36 cells (2010) [61]. These findings highlight the complexity of viral interactions within mosquito hosts and emphasize the importance of considering virus–virus, virus–vector, and cell-type-specific dynamics when investigating superinfection outcomes. Moreover, superinfection exclusion is thought to confer evolutionary advantages by balancing genetic diversification with the preservation of genomic integrity, thereby restricting recombination and the replication of defective genomes [56].

The potential for viral interference in superinfection scenarios has been extensively investigated, particularly in the context of insect-specific viruses (ISVs) [24]. ISVs replicate exclusively in insect cells and have been shown to modulate the infection, dissemination, and transmission of pathogenic arboviruses. For example, interference has been documented between Nhumirim virus (NHUV) and West Nile virus (WNV) in *Culex quinquefasciatus* [62], and between Eilat virus (EILV) and CHIKV in *Ae. aegypti* [54]. These findings underscore the potential of ISVs as tools for the development of novel vector control strategies [26]. An alternative hypothesis suggests that interviral interference among pathogenic arboviruses may partially explain the geographic and temporal exclusion patterns observed in endemic regions [24,27]. Abrao and Da Fonseca (2016) [27] proposed this mechanism to explain the relatively limited circulation of YFV in areas with a high endemicity of DENV, suggesting that viral interference may constrain the establishment or expansion of competing arboviruses. However, this hypothesis remains speculative and requires validation through integrated approaches combining in vitro experiments, in vivo models, and field-based epidemiological data.

Our findings demonstrate that, under coinfection conditions, both MAYV and CHIKV replicate in *Ae. aegypti* without significantly compromising the transmissibility of either virus. In contrast, superinfection scenarios revealed evidence of interference, with the reduced transmission efficiency of the secondarily acquired virus. These observations raise important epidemiological concerns, particularly in urban regions of Brazil where multiple arboviruses, including DENV, ZIKV, CHIKV, YFV, and OROV, co-circulate. The results underscore the necessity of implementing multiplex diagnostic strategies capable of detecting and distinguishing among multiple arboviral infections simultaneously. Furthermore, the findings highlight the complex and dynamic nature of arbovirus–vector interactions, with potential implications for transmission patterns and outbreak risk. Continued entomological and virological surveillance, combined with experimental research into arbovirus interactions, will be critical for anticipating and mitigating the public health impact of co-circulating arboviruses in endemic regions.

## 5. Conclusions

This study demonstrates that both MAYV and CHIKV replicate in *Ae. aegypti* under coinfection conditions without evident disruption to viral replication, in line with previous reports on arboviral co-infection dynamics in this vector. In contrast, superinfection scenarios revealed evidence of interference affecting the transmission of the subsequently acquired virus, reinforcing earlier studies that have documented similar interactions between CHIKV and MAYV. A striking finding of this work was the absence of detectable plaques in Vero cells despite the presence of high viral RNA titers in mosquito saliva samples. This discrepancy may reflect the production of non-infectious or defective viral particles, potentially driven by viral interference mechanisms, and underscores the importance of considering both molecular and cell culture approaches when evaluating vector competence.

These findings emphasize the need to incorporate both coinfection and superinfection dynamics into investigations of virus–vector interactions, entomological surveillance strategies, and arboviral risk assessment frameworks, particularly in regions where multiple arboviruses co-circulate.

The broader implications of these interactions, especially in the contexts of viral epidemiology, pathogenesis, and evolutionary trajectories, remain poorly understood in both mosquito vectors and vertebrate hosts. Future research should aim to clarify how coinfection and superinfection events influence viral genome diversity, intra-host competition, and the potential emergence of novel or altered pathogenic phenotypes in arbovirus-associated diseases.

## Figures and Tables

**Figure 1 microorganisms-13-02165-f001:**
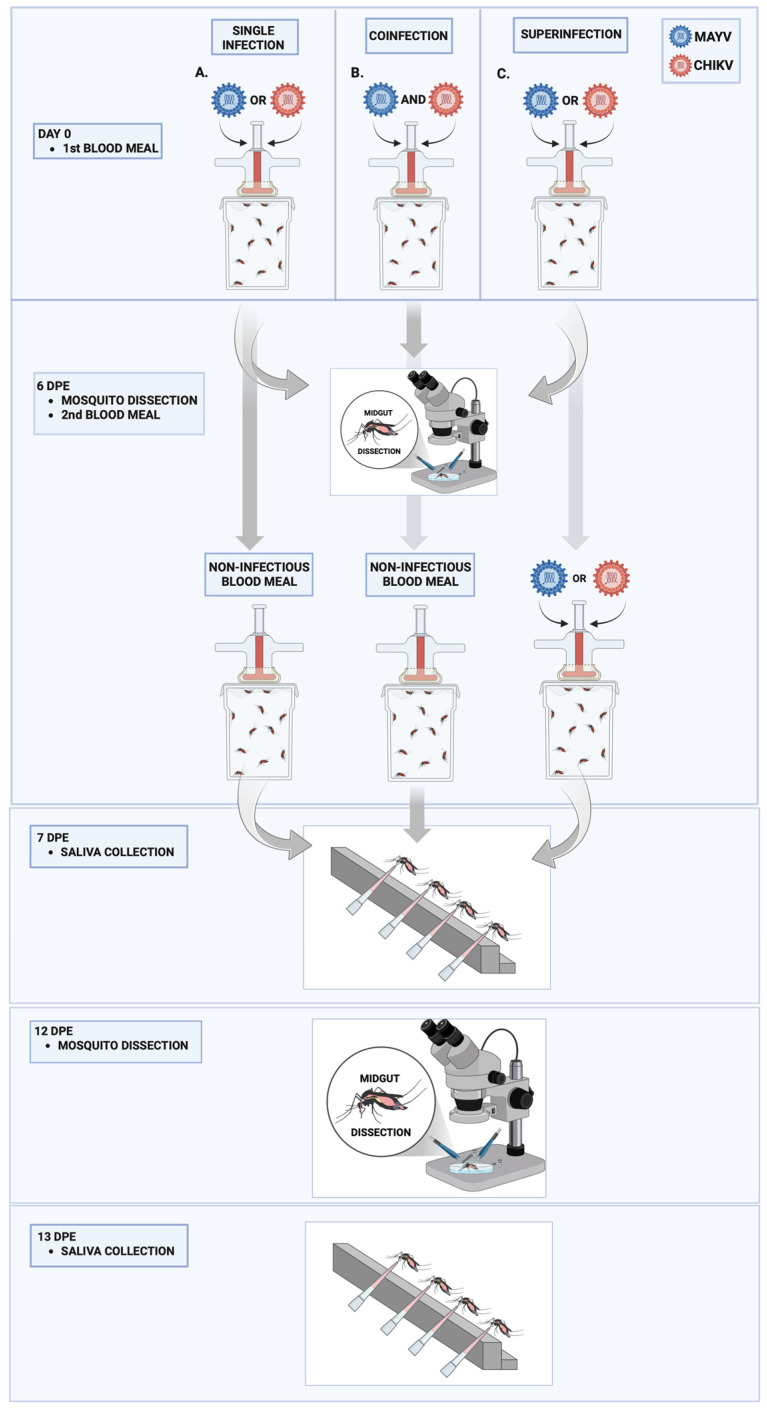
Experimental design for single infection, coinfection, and superinfection of *Aedes aegypti* with Mayaro virus (MAYV) and Chikungunya virus (CHIKV). (**A**) Single infection: Mosquitoes were fed a blood meal containing either MAYV or CHIKV on day 0; (**B**) Coinfection: Mosquitoes received a single blood meal containing both MAYV and CHIKV simultaneously on day 0; (**C**) Superinfection: Mosquitoes were first exposed to a blood meal containing either MAYV or CHIKV on day 0, followed by a second blood meal containing the heterologous virus at 6 days post-exposure (dpe). Midgut dissections were performed and a non-infectious blood meal was administered at 6 dpe. Saliva samples were collected at 7 dpe. A second round of midgut dissections and saliva collections was conducted at 12 and 13 dpe, respectively.

**Figure 2 microorganisms-13-02165-f002:**
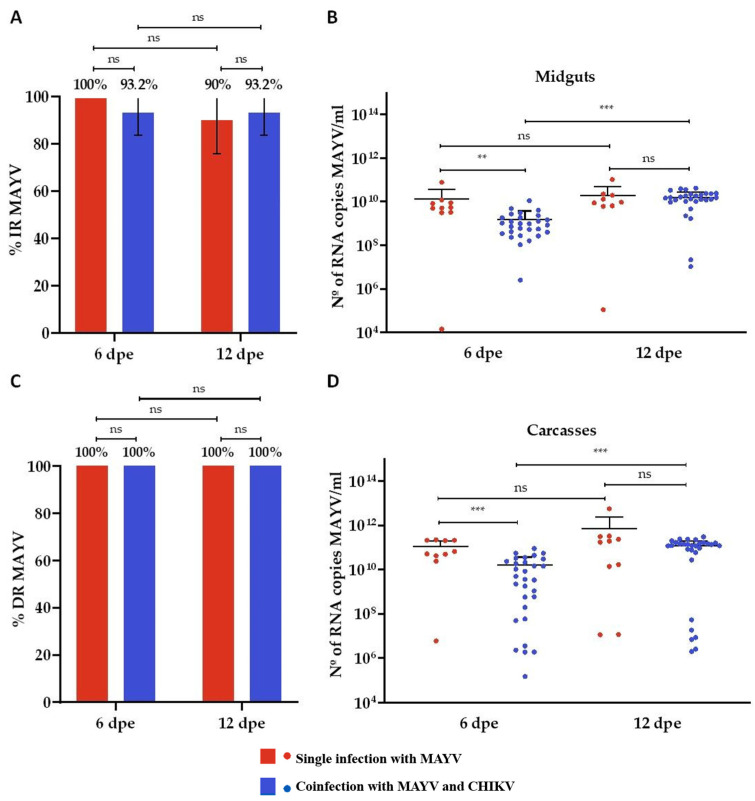
Infection and dissemination rates and viral RNA loads of Mayaro virus (MAYV) in *Aedes aegypti* following single infection and coinfection with Chikungunya virus (CHIKV). (**A**) Infection rate (IR, %) in midguts; (**B**) Viral RNA load (copies/mL) in positive midguts; (**C**) Dissemination rate (DR, %) based on carcass positivity; (**D**) Viral RNA load (copies/mL) in positive carcasses. Days post-exposure (dpe) is indicated on the *x*-axis. Statistical significance is denoted as follows: *p* ≤ 0.01 (**), *p* ≤ 0.001 (***), and ns = not significant. Error bars represent the mean ± standard deviation (SD). For each time point, *n* = 10 tissues (midguts and carcasses) were analyzed in the MAYV single-infection group, and *n* = 30 in the CHIKV–MAYV coinfection group.

**Figure 3 microorganisms-13-02165-f003:**
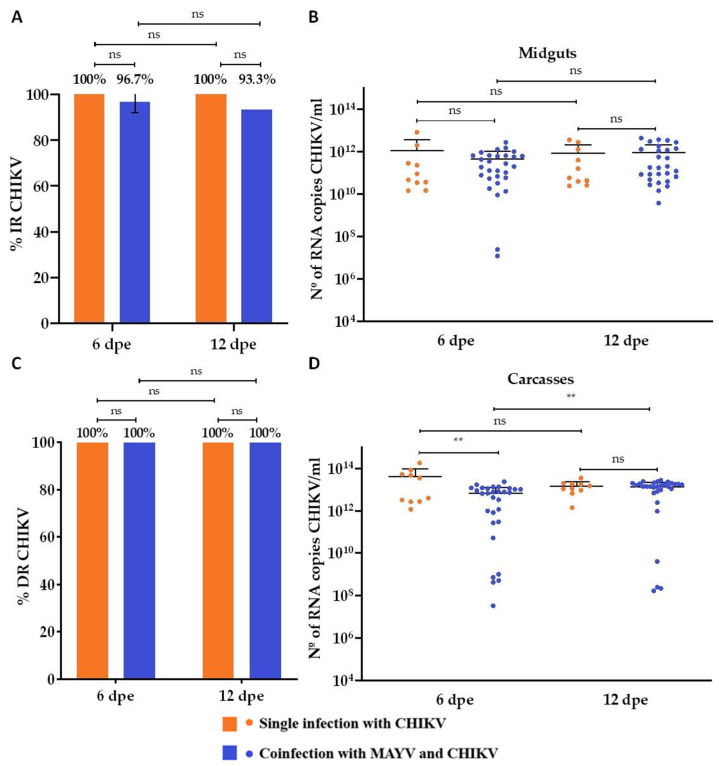
Infection and dissemination rates and viral RNA loads of Chikungunya virus (CHIKV) in *Aedes aegypti* following single infection and coinfection with Mayaro virus (MAYV). (**A**) Infection rate (IR, %) in midguts; (**B**) Viral RNA load (copies/mL) in CHIKV-positive midguts; (**C**) Dissemination rate (DR, %) based on carcass positivity; (**D**) Viral RNA load (copies/mL) in CHIKV-positive carcasses. Days post-exposure (dpe) is indicated on the *x*-axis. Statistical significance is denoted as follows: *p* ≤ 0.01 (**), and ns = not significant. Error bars represent the mean ± standard deviation (SD). For each time point, *n* = 10 tissues (midguts and carcasses) were analyzed in the CHIKV single-infection group and *n* = 30 in the CHIKV–MAYV coinfection group.

**Figure 4 microorganisms-13-02165-f004:**
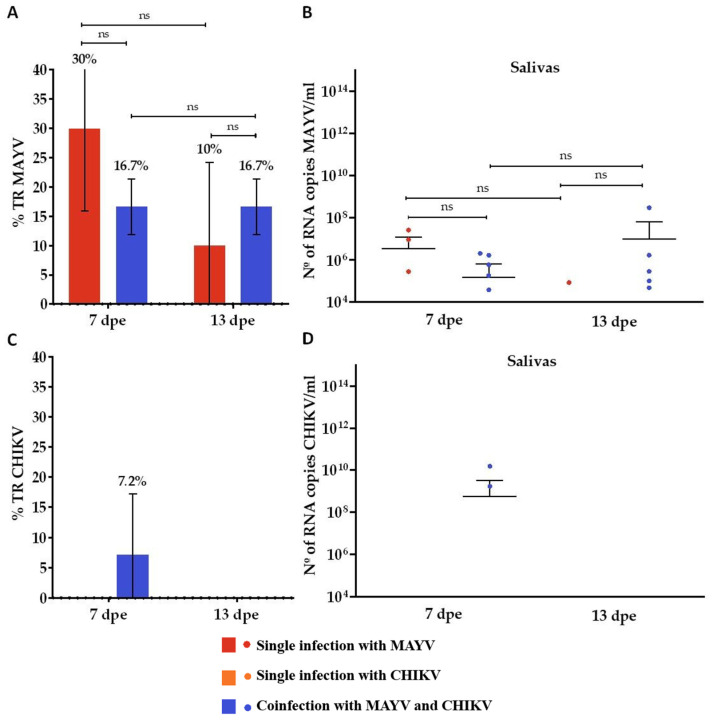
Transmission rates and viral RNA loads of Mayaro virus (MAYV) and Chikungunya virus (CHIKV) in *Aedes aegypti* following single infection and coinfection. (**A**) Transmission rate (TR, %) for MAYV; (**B**) Viral RNA load (copies/mL) in MAYV-positive saliva samples; (**C**) Transmission rate (TR, %) for CHIKV; (**D**) Viral RNA load (copies/mL) in CHIKV-positive saliva samples. Days post-exposure (dpe) is indicated on the *x*-axis. Statistical significance is represented as follows: ns = not significant. Error bars represent the mean ± standard deviation (SD). For each time point, *n* = 10 saliva samples were analyzed in the MAYV and CHIKV single-infection groups and *n* = 30 in the CHIKV–MAYV coinfection group.

**Figure 5 microorganisms-13-02165-f005:**
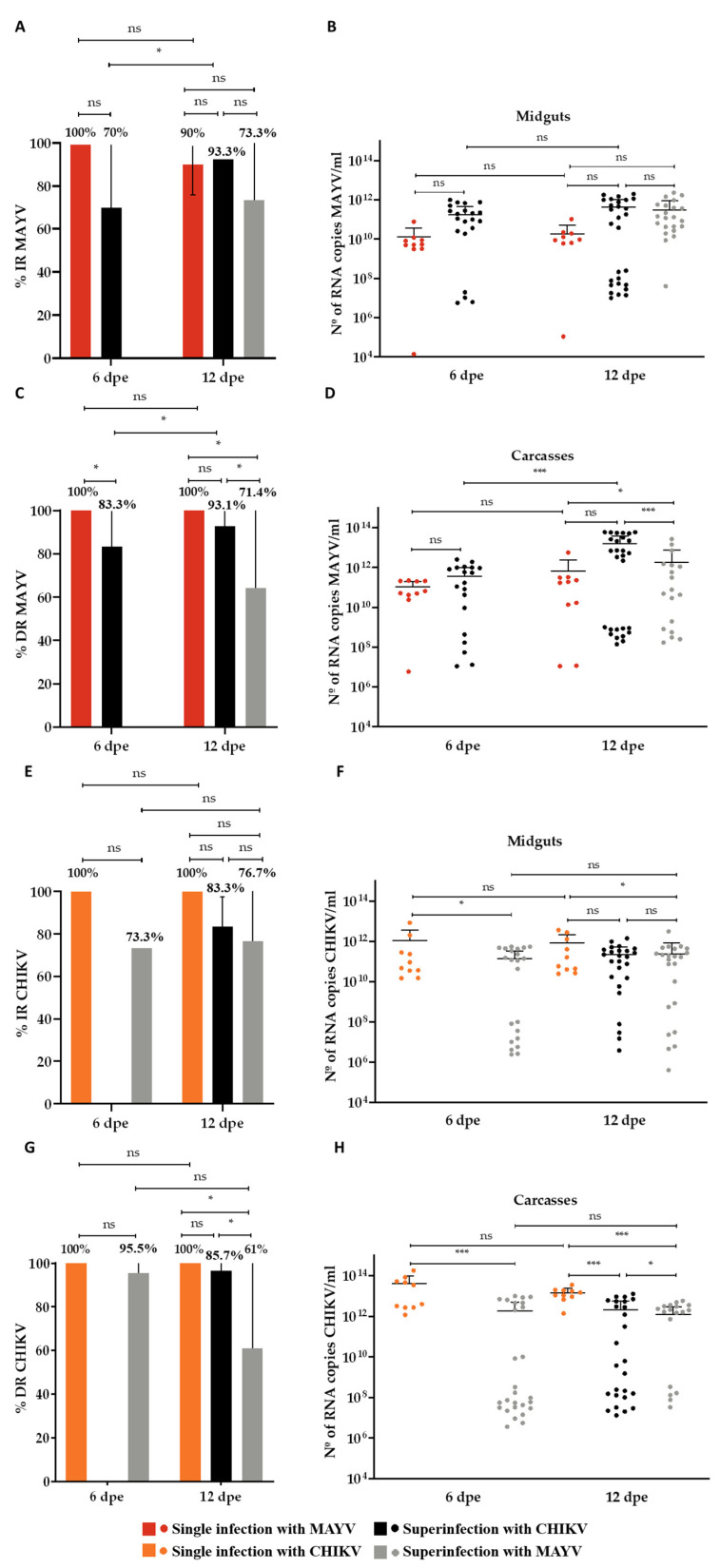
Infection and dissemination rates and viral RNA loads of Mayaro virus (MAYV) and Chikungunya virus (CHIKV) in *Aedes aegypti* following superinfection. (**A**) Infection rate (IR, %) for MAYV; (**B**) Viral RNA load (copies/mL) in MAYV-positive midguts; (**C**) Dissemination rate (DR, %) for MAYV; (**D**) Viral RNA load (copies/mL) in MAYV-positive carcasses; (**E**) Infection rate (IR, %) for CHIKV; (**F**) Viral RNA load (copies/mL) in CHIKV-positive midguts; (**G**) Dissemination rate (DR, %) for CHIKV; (**H**) Viral RNA load (copies/mL) in CHIKV-positive carcasses. Days post-exposure (dpe) is indicated on the *x*-axis. Statistical significance is indicated as follows: *p* ≤ 0.05 (*), *p* ≤ 0.001 (***), and ns = not significant. Error bars represent the mean ± standard deviation (SD). For each time point, *n* = 10 midgut and carcass samples were analyzed in the CHIKV and MAYV single-infection groups, and *n* = 30 in the respective superinfection groups.

**Figure 6 microorganisms-13-02165-f006:**
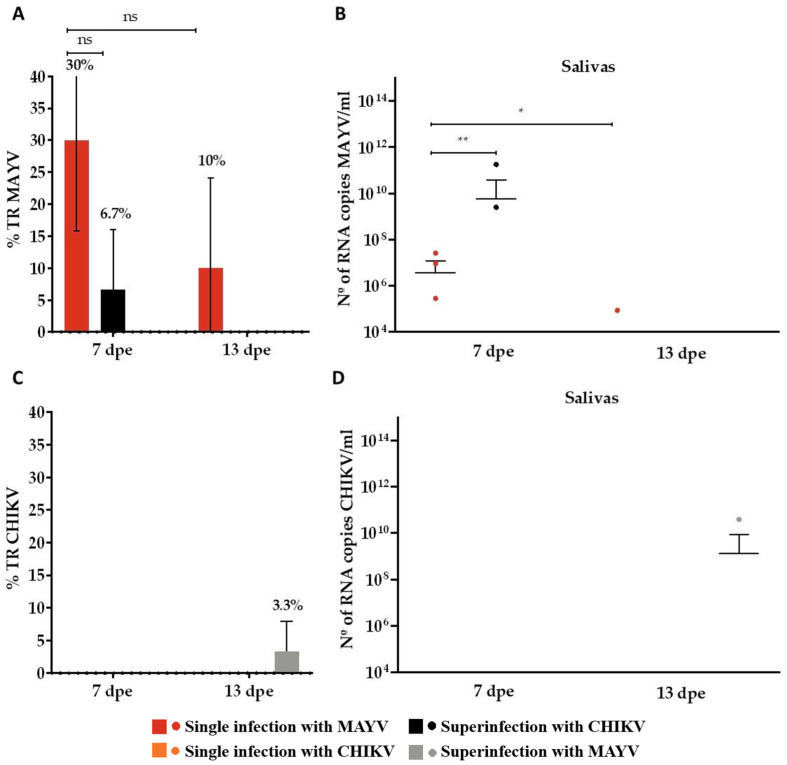
Transmission rate (%) for MAYV and CHIKV and number of MAYV and CHIKV RNA copies/mL in the superinfection assay. (**A**) Transmission rate (TR) for MAYV; (**B**) MAYV RNA copy number/mL in positive saliva; (**C**) Transmission rate (TR) for CHIKV; (**D**) CHIKV RNA copy number/mL in positive saliva. dpe: days post-exposure; * *p* ≤ 0.05, ** *p* ≤ 0.01, and ns—not significant; the error bars represent the mean ± standard deviation of the values. We analyzed *n* = 10 saliva samples for MAYV and CHIKV single-infected groups and *n* = 30 for MAYV- or CHIKV-superinfected groups.

**Table 1 microorganisms-13-02165-t001:** Average initial viral titers, expressed in plaque-forming units per milliliter (PFU/mL), for Mayaro virus (MAYV) and Chikungunya virus (CHIKV) used in single infection, coinfection, and superinfection assays. Titers were determined from infected cell culture supernatants prior to each blood-feeding experiment to ensure consistency across all experimental conditions.

Group	Virus	1st Blood Meal	2nd Blood Meal
Single infection—MAYV	MAYV	5.5 × 10^7^	- *
Single infection—CHIKV	CHIKV	3.1 × 10^7^	- *
Coinfection MAYV + CHIKV	MAYV	5.5 × 10^7^	- *
	CHIKV	3.1 × 10^7^	- *
Superinfection by MAYV	CHIKV	1.4 × 10^7^	-
	MAYV	-	3.1 × 10^7^
Superinfection by CHIKV	MAYV	3.5 × 10^7^	-
	CHIKV	-	8.5 × 10^6^

* non-infectious blood meal.

## Data Availability

The original contributions presented in this study are included in the article. Further inquiries can be directed to the corresponding author.

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
