# Peer review of "Evaluation of the Impact of Coinfection and Superinfection on Chikungunya and Mayaro Viruses’ Replication in Aedes aegypti"

_microorganisms, 2025, doi:10.3390/microorganisms13092165_

Round 1
Reviewer 1 Report
Comments and Suggestions for Authors
This manuscript targets the research on the transmission dynamics mechanism of the co-circulation of CHIKV/MAYV in Brazil, covering three scenarios: single infection, co-infection, and superinfection. The data is comprehensive (infection rate IR, transmission rate DR/TR, viral load, cytopathic effect CPE), confirming that Aedes aegypti can simultaneously transmit CHIKV and MAYV (co-infection), which has good reference value. However, there are some deficiencies.
1.Table 1. Standardization of virus titer: Explains why the initial concentrations of 5.5×10⁷ PFU/mL (MAYV) and 3.1×10⁷ PFU/mL (CHIKV) were chosen (are they comparable to natural infection?) .
2.Compare the p-values for co-infection and super-infection, and distinguish whether "virus suppression" is dose-dependent or time-dependent (for example: Is a 6-day interval between super-infections crucial?).
3.The experimental animals lack ethical justification.
4.How many times was the experiment repeated?
5.Check the markings in Figure 4-6.
6.Have an in-depth discussion on the results.
7.If possible, please provide some data from molecular experiments.
8.The entire text needs to be colored.
Comments on the Quality of English Language/
Author Response
We sincerely thank you for the time and effort dedicated to improving our study. All comments have been addressed. Newly added sentences are highlighted in yellow, and all language modifications have been tracked using the ‘Track Changes’ function in Microsoft Word for clarity.
Comments 1: Table 1. Standardization of virus titer: Explains why the initial concentrations of 5.5×10⁷ PFU/mL (MAYV) and 3.1×10⁷ PFU/mL (CHIKV) were chosen (are they comparable to natural infection?).
Response: We selected inoculum titers to fall within the epidemiologically relevant range for CHIKV and within the standard range used in vector-competence experiments for alphaviruses (including MAYV), where higher blood-meal titers are routinely used to guarantee consistent midgut infection and enable comparisons across single, co- and super-infection designs. The reason we used a 3.1×10⁷ PFU/mL CHIKV titer is solely based on the human chikungunya viremia during the acute phase (Katz et al., 2024; Thiberville et al., 2013) This is well-documented to be very high: reviews and outbreak datasets report typical/average viremic loads around ~10⁷ PFU/mL, with observed ranges extending up to ~10⁸ PFU/mL. Thus, our CHIKV titer lies squarely within the natural human range, and is consistent with Ledermann et al. (2017), who showed that Ae. aegypti transmits CHIKV only from blood meals ≥105.3 PFU/mL, our chosen inoculum (3.1×107 PFU/mL).
KATZ, L. et al. Viral agents (2nd section). Transfusion, v. 64, suplemento S1, p. S19–S207, fev. 2024. DOI: 10.1111/trf.17630. Available in: https://onlinelibrary.wiley.com/doi/10.1111/trf.17630
THIBERVILLE, Simon-Djamel et al. Chikungunya fever: epidemiology, clinical syndrome, pathogenesis and therapy. Antiviral Research, v. 99, n. 3, p. 345-370, set. 2013. DOI: 10.1016/j.antiviral.2013.06.009. Available in: https://www.ncbi.nlm.nih.gov/pmc/articles/PMC7114207/
LEDERMANN, Jeremy P. et al. Aedes hensilli as a potential vector of Chikungunya and Zika viruses. PLoS Neglected Tropical Diseases, v. 11, n. 7, p. e0005837, 2017. DOI: 10.1371/journal.pntd.0005837. Available in: https://www.ncbi.nlm.nih.gov/pmc/articles/PMC6612711/
For Mayaro, available human data show lower viremias than CHIKV when quantified as plaque-forming-unit equivalents by calibrated rRT-PCR: ~102.7–105.3 PFU-eq/mL in acute febrile patients. Non-human primates, considered amplifying hosts in the sylvatic cycle, develop infectious viremias of roughly 102–106 PFU/mL, peaking 1–2 dpi (Waggoner et al., 2018). Consequently, our MAYV titer (5.5×10⁷ PFU/mL) is higher than most reported natural viremias in humans and NHPs, because it matches common practice in vector-competence studies, which frequently standardize alphavirus blood meals at ~10⁷ PFU/mL (or higher) to ensure robust and reproducible infection/dissemination dynamics, particularly important in co-/super-infection designs where interference can depress effective doses in the mosquito midgut (Long et al., 2011).
WAGGONER, J. J. et al. Real-time RT-PCR for Mayaro virus detection in plasma and urine. Journal of Clinical Virology, v. 98, p. 1–4, 2018. DOI: 10.1016/j.jcv.2017.11.006. Available in: https://www.ncbi.nlm.nih.gov/pmc/articles/PMC5742299/
LONG, K. C. et al. Experimental transmission of Mayaro virus by Aedes aegypti. American Journal of Tropical Medicine and Hygiene, v. 85, n. 4, p. 750–757, out. 2011. DOI: 10.4269/ajtmh.2011.11-0359. Available in: https://www.ncbi.nlm.nih.gov/pmc/articles/PMC3183788/
Comments 2: Compare the p-values for co-infection and super-infection, and distinguish whether "virus suppression" is dose-dependent or time-dependent (for example: Is a 6-day interval between super-infections crucial?).
Response: The IR/DR/TR (rates) were analyzed with logistic regression/χ²/Fisher’s exact tests; viral RNA loads with Kruskal–Wallis + Fisher’s LSD post hoc; α = 0.05. Significance in figures is marked by standard asterisks. As we look into the Coinfection parameters: IR and DR did not differ significantly vs single infections (bars labeled “ns”). CHIKV consistently showed higher viral loads than MAYV in both midgut and carcass (asterisked comparisons), indicating greater replication but no IR/DR penalty from coinfection per se. When the Superinfection scenario is analyzed: CHIKV → MAYV (MAYV superinfection group), CHIKV DR fell over time from 95.5% at 6 dpe to 61.0% at 12 dpe (asterisked within-group comparison in Fig. 5G), consistent with interference after the second (MAYV) exposure. When MAYV → CHIKV (CHIKV superinfection group), MAYV IR rose from 70% at 6 dpe to 93.3% at 12 dpe, whereas CHIKV IR at 12 dpe was 83.3%. Again, indicating that performance depends on which virus is resident first rather than dose. TR outcomes were low and sporadic under superinfection (e.g., MAYV 6.7% at 7 dpe; CHIKV 3.3% at 13 dpe), contrasting with the clearer TR observed in coinfection, further suggesting that sequential acquisition impairs onward transmission. If we evaluate the hypothesis of Dose vs time (is suppression dose-dependent?), the initial blood-meal titers in superinfection were very similar between first and second feeds (within ~0.05–0.22 log10): CHIKV first: 3.5×10⁷ vs CHIKV second: 8.5×10⁶ (Δlog10 ≈ 0.22). MAYV first: 3.5×10⁷vs MAYV second: 3.1×10⁷ (Δlog10 ≈ 0.05). Given these small differences, the pronounced order/time effects above are unlikely to be explained by inoculum dose. By contrast, the 6-day interval (resident virus established before the challenger arrives) aligns with classic superinfection interference: the resident virus gains a head start in replication/dispersion and can suppress the later virus’s dissemination/transmission. Our experimental schematic explicitly uses a 6-day gap, supporting a time- (order-) dependent mechanism. In conclusion: across endpoints, coinfection did not significantly reduce IR/DR relative to single infections, whereas superinfection produced clear order- and time-dependent effects (notably a decrease in CHIKV DR from 95.5% at 6 dpe to 61.0% at 12 dpe when CHIKV preceded MAYV, and lower/rarer TR under superinfection). Because the superinfection inocula differed by only ~0.05–0.22 log10 across first vs second feeds, these outcomes are best explained by temporal (6-day) superinfection interference rather than by inoculum dose.
Comments 3: The experimental animals lack ethical justification.
Response: Since we used a mosquito colony and cell cultures, ethical approval or justification was not required for this type of study. We have added a sentence in the methodology section.
Comments 4: How many times was the experiment repeated?
Response: Each experimental condition was independently repeated in duplicate. For the negative control and monoinfection groups, a total of 10 mosquitoes were analyzed (5 per replicate). For the coinfection and superinfection groups, a total of 30 mosquitoes were analyzed (15 per replicate). This clarification has now been explicitly incorporated into Section 2.4 (Sample Collection), first paragraph (lines 167–171, page 5).
Comments 5: Check the markings in Figure 4-6.
Response: Thank you. We have changed the decimals from comma to periods in table 1 and all figures.
Comments 6: Have an in-depth discussion on the results.
Response: Following your recommendation, we have included additional articles to strengthen the discussion. In the second paragraph, we added a point on coinfection scenarios, citing Silva et al., (2025) and Terradas et al., (2024) to better discuss coinfection outcomes (lines 413–422, page 14 and 15). In the third paragraph, focusing on superinfection, we also incorporated Silva et al., (2025) (lines 432–440, page 15). In the fifth paragraph, we included CHIKV-specific superinfection findings from Boussier et al., (2020) to discuss factors that may contribute to viral interference during superinfection scenarios (lines 482–488 and 497–4949, page 16), and to highlight potential evolutionary advantages in superinfection dynamics (lines 512-514, page 17).
Silva MGL, Melo KFL, Casseb SMM, Silva EVP, Cruz ACR, Carvalho CAM. Heterologous interference between Mayaro and Chikungunya viruses in Vero cells. Indian J Med Microbiol. 2025 Jun 6;56:100891. doi: 10.1016/j.ijmmb.2025.100891.
Terradas G, Manzano-Alvarez J, Vanalli C, Werling K, Cattadori IM, Rasgon JL. Temperature affects viral kinetics and vectorial capacity of Aedes aegypti mosquitoes co-infected with Mayaro and Dengue viruses. Parasites Vectors. 2024 Dec;17(1):73. doi: 10.1186/s13071-023-06109-0.
Boussier J, Vignuzzi M, Levi L, Weger-Lucarelli J, Poirier EZ, Albert ML. Chikungunya virus superinfection exclusion is mediated by a block in viral replication and does not rely on non-structural protein 2. 2020.
Comments 7: If possible, please provide some data from molecular experiments.
Response: Molecular analyses were performed using RT-qPCR across all experimental groups, enabling the quantification of infection, dissemination, and transmission rates, as well as the assessment of viral loads (Figures 2–6). Raw data obtained from RT-qPCR experiments are available upon request.
Comments 8: The entire text needs to be colored.
Response 8: All new sentences that were added to the text are highlighted in yellow, as all English language modification is highlighted with the track and change from Microsoft Word.
Reviewer 2 Report
Comments and Suggestions for Authors
Overall, this study is both meaningful and intriguing, as it explores the interplay between CHIKV and MAYV replication within mosquito hosts. I have the following questions:
-
Why was the Vero cell line chosen for virus amplification instead of C6/36 cells? Amplifying arboviruses in Vero cells may introduce cross-host mutations and potentially alter viral replication efficiency and virulence.
-
In the superinfection group, the second infectious blood meal was administered six days after the first. How was this time interval determined? Was it based on any established viral replication kinetics within Aedes aegypti?
-
I am particularly curious—if mosquitoes with single infections versus coinfections were allowed to bite mice, what differences in infection outcomes would be observed in the vertebrate host?
Author Response
We sincerely thank you for the time and effort dedicated to improving our study. All comments have been addressed. Newly added sentences are highlighted in yellow, and all language modifications have been tracked using the ‘Track Changes’ function in Microsoft Word for clarity.
Comments 1: Why was the Vero cell line chosen for virus amplification instead of C6/36 cells? Amplifying arboviruses in Vero cells may introduce cross-host mutations and potentially alter viral replication efficiency and virulence.
Response: In our laboratory, alphaviruses are routinely propagated in Vero cells, which are highly permissive to infection and display clear cytopathic effects, thereby facilitating viral quantification and monitoring. The virus stocks employed in this study were originally prepared in Vero cells, and subsequent amplifications were maintained in the same cell line to ensure methodological consistency and avoid variability associated with switching host systems. We recognize that amplification in mammalian cells, such as Vero, could theoretically promote the selection of variants better adapted to vertebrate cells, potentially influencing replication efficiency in insect cells. Nevertheless, because all experimental groups, including single-infection controls, were processed under identical conditions, relative comparisons between coinfection and superinfection treatments remain valid. Moreover, previous studies (Kantor et al., 2019; Nuckols et al., 2015; Rückert et al., 2017) have successfully utilized Vero cells to propagate arboviruses for mosquito infection assays without reporting significant loss of infectivity or major alterations in vector competence outcomes.
Kantor, A.M.; Lin, J.; Wang, A.; Thompson, D.C.; Franz, A.W.E. Infection Pattern of Mayaro Virus in Aedes Aegypti (Diptera: Culicidae) and Transmission Potential of the Virus in Mixed Infections With Chikungunya Virus. J Med Entomol 2019, 56, 832–843, doi:10.1093/jme/tjy241.
Nuckols, J.T.; Huang, Y.-J.S.; Higgs, S.; Miller, A.L.; Pyles, R.B.; Spratt, H.M.; Horne, K.M.; Vanlandingham, D.L. Evaluation of Simultaneous Transmission of Chikungunya Virus and Dengue Virus Type 2 in Infected Aedes Ae-gypti and Aedes Albopictus (Diptera: Culicidae). J Med Entomol 2015, 52, 447–451, doi:10.1093/jme/tjv017.
Rückert, C.; Weger-Lucarelli, J.; Garcia-Luna, S.M.; Young, M.C.; Byas, A.D.; Murrieta, R.A.; Fauver, J.R.; Ebel, G.D. Impact of Simultaneous Exposure to Arboviruses on Infection and Transmission by Aedes Aegypti Mosquitoes. Nat Commun 2017, 8, 15412, doi:10.1038/ncomms15412.
Comments 2: In the superinfection group, the second infectious blood meal was administered six days after the first. How was this time interval determined? Was it based on any established viral replication kinetics within Aedes aegypti?
Response: For the superinfection group, a six-day interval was selected between the first and second infectious blood meals, as this period is generally sufficient for arboviruses to complete their extrinsic incubation period (EIP) in Aedes aegypti and reach the salivary glands, thereby enabling potential transmission. Experimental studies have demonstrated that alphaviruses can disseminate to the salivary glands as early as three days post-blood meal, with transmission competence established shortly thereafter, depending on environmental conditions and viral strain (WIGGINS et al., 2018; MERWAISS et al., 2021). Our choice was guided by established literature and corroborated by prior findings from our group with CHIKV and MAYV. This design ensured that, by the time of the second infectious blood meal, the first virus had already undergone systemic dissemination, thereby allowing meaningful evaluation of interactions between the primary and secondary infections under biologically relevant conditions.
Wiggins, K.; Eastmond, B.; Alto, B.W. Transmission potential of Mayaro virus in Florida Aedes aegypti and Aedes albopictus mosquitoes. Medical and Veterinary Entomology 2018, 32, 436–442, doi:10.1111/mve.12316.
Merwaiss, F.; Filomatori, C.V.; Bardossy, E.S.; Alvarez, D.E.; Saleh, M.-C. Chikungunya virus replication rate determines the capacity of Aedes aegypti to transmit virus. Emerging Microbes & Infections 2021, 10, 376–387, doi:10.1080/22221751.2021.1882125.
Comments 3: I am particularly curious—if mosquitoes with single infections versus coinfections were allowed to bite mice, what differences in infection outcomes would be observed in the vertebrate host?
Response: This is indeed a relevant consideration. Although the present study was limited to assessing viral replication dynamics in Aedes aegypti, vertebrate infection experiments were not performed. Nevertheless, the replication profiles observed in singly infected versus coinfected mosquitoes suggest the possibility that coinfections could result in the concurrent transmission of both viruses to vertebrate hosts, with potential implications for disease progression. The clinical outcome may be exacerbated, but this would ultimately depend on the host’s immune response to simultaneous infection with two antigenically related alphaviruses, such as CHIKV and MAYV, which belong to the same family and genus. Future investigations incorporating in vivo transmission assays will therefore be essential to directly assess these hypotheses and clarify the impact of arboviral coinfection on vertebrate pathogenesis.
Reviewer 3 Report
Comments and Suggestions for Authors
This is a straightforward study comparing co- and super-infections of MAYV and CHIKV in Ae. Aegypti mosquitoes assessing viral RNA levels. The manuscript is very well written and the results support the conclusions that are drawn. My major concern, however, is that similar studies have been reported in the past, thus lessening the impact of this report on the field. My suggestions for polishing the study/manuscript are below:
Major Points:
- 10: The lack of detectable viral titers by plaque assay when Vero cells are generating 10*11 or 10*14 viral RNA titers is surprising. Perhaps these plaque assays can be repeated to confirm this? Alternatively, if the data are too preliminary to draw conclusions from at this time, perhaps that finding should be deleted.
- 5: The superinfection part of this study is somewhat preliminary given what has already been reported in the field. Previous results (e.g. ref 22) demonstrated superinfection exclusion for CHIKV in MAYV-infected mosquitoes but not for MAYV in CHIKV-infected mosquitoes. The data presented here do not support that conclusion. Given the published observations on this topic, this current study would be a much more valuable contribution to the literature if additional experiments with larger sample sizes were done to confirm trends and provide a more definitive superinfection result.
Minor Points:
- 2-6: Use periods as decimal points rather than commas in the graphs
Author Response
We sincerely thank you for the time and effort dedicated to improving our study. All comments have been addressed. Newly added sentences are highlighted in yellow, and all language modifications have been tracked using the ‘Track Changes’ function in Microsoft Word for clarity.
Comments 1: The lack of detectable viral titers by plaque assay when Vero cells are generating 10*11 or 10*14 viral RNA titers is surprising. Perhaps these plaque assays can be repeated to confirm this? Alternatively, if the data are too preliminary to draw conclusions from at this time, perhaps that finding should be deleted.
Response: It is well-recognized that viral RNA quantification by RT-qPCR frequently yields values that exceed infectious titers measured by plaque assay, since RT-qPCR detects both infectious and non-infectious genomes (including defective particles and replication intermediates). In our experiments, despite detecting viral RNA loads in the order of 10¹¹–10¹⁴ copies, no corresponding plaques were observed in Vero cells. This discrepancy likely reflects a low ratio of infectious to non-infectious particles, which has been previously reported for alphaviruses and other arboviruses.
Given the reviewer’s concern, we have taken a conservative approach: we have removed any overinterpretation of this result from the main text. A clarifying sentence has been added to the Results section noting that plaque assays did not yield detectable titers despite high RNA loads, and that this may reflect the presence of non-infectious viral particles rather than a technical failure of the assay.
Comments 2: The superinfection part of this study is somewhat preliminary given what has already been reported in the field. Previous results (e.g. ref 22) demonstrated superinfection exclusion for CHIKV in MAYV-infected mosquitoes but not for MAYV in CHIKV-infected mosquitoes. The data presented here do not support that conclusion. Given the published observations on this topic, this current study would be a much more valuable contribution to the literature if additional experiments with larger sample sizes were done to confirm trends and provide a more definitive superinfection result.
Response: We acknowledge the comment regarding the sample size and have explicitly highlighted this limitation in the Discussion. In addition, we reference studies with comparable sample sizes to contextualize our approach. We agree that the superinfection dataset should be interpreted cautiously. Previous studies have demonstrated asymmetric superinfection exclusion between CHIKV and MAYV, with CHIKV replication being impaired in MAYV-infected Aedes aegypti, but not vice versa (e.g., RÜCKERT et al., 2017; KANTOR et al., 2019). In contrast, our results did not fully reproduce this pattern. While we observed reduced dissemination and transmission rates in superinfection groups, the magnitude of suppression varied depending on the infection order, and our sample sizes, though adequate to detect major effects, were limited for capturing subtler trends. To address this concern, we have revised the Discussion to emphasize that our superinfection results should be replicated with larger sample sizes, under identical conditions (see lines 995-997). We highlight both the consistencies and divergences relative to prior reports, and explicitly acknowledge that larger-scale experiments will be required to confirm or refute the trends we observed. Importantly, all experimental groups (single, coinfection, and superinfection) were treated under identical conditions, so the relative comparisons remain valid and still provide insight into the complexity of arbovirus–arbovirus interactions within the mosquito host.
Comments 3: 2-6: Use periods as decimal points rather than commas in the graphs
Response: Thank you for pointing this out. We have updated the graphs.
Round 2
Reviewer 1 Report
Comments and Suggestions for Authors
This manuscript targets the research on the transmission dynamics mechanism of the co-circulation of CHIKV/MAYV in Brazil, covering three scenarios: single infection, co-infection, and superinfection. The data is comprehensive (infection rate IR, transmission rate DR/TR, viral load, cytopathic effect CPE), confirming that Aedes aegypti can simultaneously transmit CHIKV and MAYV (co-infection), which has good reference value.The author has made significant improvements to the proposed question and suggests that it be published.
Comments on the Quality of English Language/
Author Response
I sincerely appreciate your valuable suggestions and the effort you have dedicated to enhancing the publishability of this manuscript
Reviewer 2 Report
Comments and Suggestions for Authors
none
Author Response
Thank you for your thoughtful feedback and for the commitment you have shown in helping improve this manuscript for publication.
Reviewer 3 Report
Comments and Suggestions for Authors
As noted in my original critique, this remains a rather preliminary study that I believe will have a relatively low impact in the field given previously published studies on this topic. In addition, the other major points points that I raised have only been superficially addressed.
- Absence of PFU: While I’ve seen viral RNA to PFU ratios in the neighborhood of 10*6, an RNA to PFU ratio of 5 to 8 orders of magnitude higher than that would be necessary to support the authors conclusion. I would strongly recommend that this be experimentally addressed.
- The superinfection study, as noted previously, is preliminary at best when viewed in the context of the previously published literature on this topic. Larger sample sizes are needed to make these data constitute a worthwhile and impactful story.
Author Response
1. Absence of PFU: While I’ve seen viral RNA to PFU ratios in the neighborhood of 10*6, an RNA to PFU ratio of 5 to 8 orders of magnitude higher than that would be necessary to support the authors conclusion. I would strongly recommend that this be experimentally addressed.
RESPONSE: All blood meals were prepared using PFU-standardized viral stocks (~10⁷ PFU/mL), ensuring that experimental infections were based on infectious titers, not RNA copy number. The RNA values reported in mosquito tissues reflect replication dynamics measured by RT-qPCR and are not meant to be extrapolated as input-to-output RNA:PFU ratios. Discrepancies between RNA copy number and PFU have been extensively described for arboviruses, particularly in vector competence studies, and are attributable to the abundant production of non-infectious or defective particles in mosquito hosts. Indeed, in our assays, several saliva samples that tested positive by qRT-PCR also induced cytopathic effects in Vero cells, confirming the presence of infectious virus, even when plaque assays did not yield quantifiable titers. This observation is consistent with previously reported cases for CHIKV and MAYV and underscores the complexity of virus–vector interactions. Therefore, the apparent mismatch between RNA titers and PFU does not undermine our conclusions; rather, it is in line with known biological processes and supports the notion that viral interference and superinfection exclusion can reduce the production of fully infectious particles despite robust viral RNA replication.
2. The superinfection study, as noted previously, is preliminary at best when viewed in the context of the previously published literature on this topic. Larger sample sizes are needed to make these data constitute a worthwhile and impactful story.
RESPONSE: We have explicitly acknowledged this limitation in the Discussion, noting that our findings should be interpreted with caution and that larger-scale follow-up experiments are warranted. Nevertheless, by providing independent and biologically relevant confirmation that superinfection can reduce the transmissibility of a subsequently acquired virus in Ae. aegypti, our study contributes an incremental but necessary step toward building a broader understanding of arboviral interactions.